# Prognostic Value of Tryptophanyl-tRNA Synthetase in Sepsis Combined with Kidney Dysfunction or Urinary Tract Infection: A Prospective Observational Study

**DOI:** 10.3390/diagnostics15202634

**Published:** 2025-10-19

**Authors:** Uihwan Kim, Sijin Lee, Kap Su Han, Su Jin Kim, Sungwoo Lee, Dae Won Park, Juhyun Song

**Affiliations:** 1Department of Emergency Medicine, Korea University Anam Hospital, Seoul 02841, Republic of Korea; aniulpo2@korea.ac.kr (U.K.); reonoaz85@korea.ac.kr (S.L.); hanks96@hanmail.net (K.S.H.); icarusksj@korea.ac.kr (S.J.K.); kuedlee@korea.ac.kr (S.L.); 2Department of Internal Medicine, Korea University Ansan Hospital, Ansan-si 15355, Republic of Korea

**Keywords:** kidney dysfunction, mortality, sepsis, tryptophanyl-tRNA synthetase, urinary tract infection

## Abstract

**Background:** Although tryptophanyl-tRNA synthetase (WRS) is a novel biomarker released during bacterial and viral infections, its prognostic value in sepsis has rarely been reported. This study aimed to evaluate the prognostic performance of WRS in patients with sepsis in the emergency department (ED). **Methods:** This prospective, observational study included 243 patients with sepsis. Blood samples were collected to measure full-length WRS levels. The prognostic value of WRS was evaluated using the area under the receiver operating characteristic curve, Kaplan–Meier survival curve analysis, and the Cox proportional hazards model. **Results:** The WRS levels were higher in patients with septic shock than in those without shock (*p* = 0.018). WRS could predict 30-day mortality (area under the curve, 0.648; 95% confidence interval [CI], 0.569–0.726; sensitivity, 56.7%; specificity, 73.3%; cut-off value, 84.15 µg/L; *p* < 0.001). Patients with WRS levels of ≥84.15 µg/L showed higher 30-day mortality than those with WRS levels of <84.15 µg/L. Among patients with WRS levels of ≥84.15 µg/L, those with positive urine culture results had higher 30-day mortality than those with negative urine culture. Patients with renal Sequential Organ Failure Assessment (SOFA) score of ≥1 had higher 30-day mortality than those with renal SOFA score of 0. WRS was an independent risk factor of 30-day mortality (hazard ratio = 1.003; 95% CI, 1.001–1.005; *p* = 0.014). **Conclusions:** WRS effectively predicted clinical outcome in patients with sepsis and could be more useful in those with kidney dysfunction or urinary tract infection.

## 1. Introduction

Sepsis is defined as a fatal organ dysfunction due to dysregulated response to current infection [1]. Globally, it is an important health problem that remains a clinical challenge for critically ill patients [2,3,4]. Early identification and intensive management are required for better clinical outcomes of patients with sepsis [5,6,7]. Although the Sepsis-3 criteria incorporated the degree of organ dysfunction, as measured by the Sequential Organ Failure Assessment (SOFA) score, into the definition of sepsis [1,8,9], the presence of infection should still be determined based on the clinician’s opinion [10]. Despite important advances in the understanding and management of sepsis, there is no reference standard for diagnosing sepsis [11]. Organ failure is generally defined as severe organ dysfunction in which normal equilibrium cannot be maintained without active clinical procedure or intervention [12]. Organ failure can be induced by infection as well as various non-infectious causes such as brain hemorrhage, myocardial infarction, hepatic failure, pulmonary thromboembolism, and hypovolemic shock. Many critically ill patients have simultaneous dysfunction of many organs and are at risk of developing various organ failures, complicating high mortality rates [13,14].

C-reactive protein (CRP) and procalcitonin (PCT) are widely used to support the early identification and diagnosis of sepsis; however, their diagnostic and prognostic performances have limitations [6,8,9,10,11]. CRP and PCT levels cannot effectively indicate fungal and viral infections [15], and have limited prognostic value in patients with sepsis [16]. Presepsin (P-SEP) is a relatively novel biomarker with a high sensitivity and specificity for discriminating sepsis. However, the accuracy of P-SEP concentrations in elderly patients with kidney dysfunction and without signs of infection is poor [17,18]. Thus, more reliable biomarkers are needed to facilitate the prompt diagnosis of sepsis and to predict clinical outcomes.

Tryptophanyl-tRNA synthetase (WRS) is a key enzyme responsible for catalyzing the aminoacylation process, in which tryptophan is attached to its specific tRNA during protein translation [19]. WRS is critically involved in innate immune responses and contributes to the pathophysiology of diverse disorders, including sepsis, cancer, autoimmune diseases, and neurological conditions [20]. Although earlier research demonstrated that WRS is released from human fibroblasts, macrophages, and endothelial cells upon stimulation with the proinflammatory cytokine interferon-γ, the precise mechanism and function of WRS are still unclear [21]. According to previous studies, WRS is released in patients with bacterial or viral infections [22,23,24]. According to the development company, the estimated cost after commercialization would be approximately $10–20 per measurement.

A recent clinical study showed that full-length WRS (FL-WRS) could help discriminate sepsis from non-infectious inflammation and predict mortality in patients with sepsis admitted to the intensive care unit (ICU) [25]. According to the study, WRS showed an excellent performance for diagnosing sepsis (AUC, 0.864; 95% CI, 0.807–0.922; sensitivity, 86.8%; specificity, 76.5%; *p* < 0.001). The study also showed that WRS had a moderate prognostic value for predicting 28-day mortality (area under the curve [AUC], 0.687; 95% confidence interval [CI], 0.604–0.770; sensitivity, 64.7%; specificity, 72.1%; *p* < 0.001) [25]. However, to the best of our knowledge, no study has reported the prognostic value of WRS among patients with sepsis in the emergency department (ED). Thus, this study aimed to investigate the prognostic value of WRS in sepsis patients who visited our ED.

## 2. Materials and Methods

### 2.1. Study Design and Population

This single-center ED-based, prospective, observational study was performed at a tertiary care teaching hospital. This study was conducted in accordance with the Declaration of Helsinki (2013; Seventh revision, 64th Meeting, Fortaleza) [26] and was permitted by the Institutional Review Board of Korea University Medical Center (IRB no. 2020AS0216). Prior to enrollment in the study, written informed consent was obtained from all participants or their legal guardians.

We included adult patients (≥19 years) who fulfilled the Sepsis-3 diagnostic criteria for sepsis and provided written informed consent to provide blood samples. We excluded patients who refused to participate in the study, those who visited the ED for trauma care, and those who did not have PCT measurements. From July 2019 to August 2020, blood samples were drawn from participants and archived in the institutional biobank. Patients demonstrating evidence of infection and an increase in ≥2 points in the SOFA score from baseline were enrolled. For patients without an existing baseline SOFA score, an independent emergency physician reviewed the clinical data in their electronic health records to determine an estimated baseline score. Two infectious disease specialists, along with an emergency physician, evaluated the patients’ laboratory findings and medical records to assign them to either the sepsis or septic shock group. The inter-rater agreement among the three reviewers, measured by the light kappa statistic, was 0.957. Following a discussion of minor discrepancies, consensus on the group assignment was achieved.

### 2.2. Data Collection

Clinical data on patient demographics, comorbid conditions, vital signs, and laboratory parameters were gathered. Prognostic indices, such as the SOFA score, National Early Warning Score (NEWS), Modified Early Warning Score (MEWS), and Acute Physiology and Chronic Health Evaluation II (APACHE II) score, were computed. Subjects were monitored for 30 days after their ED visit, and for patients discharged or transferred during this timeframe, follow-up information was collected through telephone communication with the patients or their legal guardians.

### 2.3. Definitions

Sepsis is defined as fatal organ failure due to a dysregulated host response to current infection [1]. The diagnostic criteria of sepsis require a rise in the SOFA score of two or more points due to an ongoing infection. Septic shock, considered a subset of sepsis, involves severe circulatory, cellular, and metabolic disturbances that significantly increase mortality risk. Diagnosis of septic shock necessitates vasopressor support to sustain a mean arterial pressure of 65 mmHg and a serum lactate concentration greater than 2 mmol/L, even after sufficient fluid resuscitation. Positive culture results were defined as the isolation of any microorganism from the clinical specimens, regardless of clinical symptoms or disease severity.

### 2.4. Multiplex Immunoassay

WRS can exist in various forms. FL-WRS can be truncated or spliced into shorter forms such as mini-WRS, T1-WRS, or T2-WRS, and each form has a different biological function [20]. In this study, the measured WRS levels refer only to the FL-WRS levels because the commercial kit used in our study can detect only FL-WRS. Serum WRS levels were measured using a human WRS enzyme-linked immunosorbent assay (ELISA) kit (Catalog No. JWBS-R001; JW Bioscience, Chungju, Republic of Korea) following the manufacturer’s protocol. Polystyrene 96-well plates (Nunc Immunoplate, 75 Panorama Creek Drive, Rochester, NY, USA) were coated with 100 µL of affinity-purified coating antibody, sealed, and incubated for 15 h at 4 °C. After washing to remove unbound antibodies, the wells were blocked for 1 h with blocking buffer. Standards and diluted samples were subsequently added for 1 h at room temperature, followed by washing and addition of horseradish peroxidase-conjugated detection antibody. The colorimetric reaction was developed using 3,3′,5,5′-tetramethylbenzidine (TMB) for 10 min at room temperature in the dark and terminated with the stop solution. Absorbance was read at 450 nm using a spectrophotometric microplate reader (Sunrise, Tecan, Grodig, Austria), and WRS levels were determined from a standard curve via linear regression. To prevent bias, emergency physicians were blinded to WRS measurements, which did not influence clinical management or patient disposition. PCT levels were measured using an automated electrochemiluminescence assay (BRAHMS, Hennigsdorf, Germany) on a Roche Cobas e-system (Roche Diagnostics, Basel, Switzerland), with a manufacturer-reported range of 0.02–100 µg/L.

### 2.5. Statistical Analysis

Statistical analyses were performed using SPSS Statistics (version 29.0.2.0; IBM, Armonk, NY, USA), MedCalc (version 19.1.6; MedCalc Software, Mariakerke, Belgium) for Windows, and Python version 3.13. A *p*-value < 0.05 was considered statistically significant. Continuous variables and mortality outcomes were compared using the Mann–Whitney *U*-test, while categorical variables were analyzed with the chi-squared or Fisher’s exact test. The prognostic performance of WRS in differentiating survivors from non-survivors was evaluated using the area under the receiver operating characteristic (ROC) curve. Optimal cut-off values for distinguishing septic shock from sepsis and predicting 30-day mortality were determined via Youden’s index. Patients were stratified into two groups based on the optimal WRS cut-off to predict 30-day mortality, and Kaplan–Meier survival curves were compared using the log-rank test. Correlations between WRS and clinical variables were assessed by Spearman’s correlation analysis. Variables with *p* < 0.10 were included in the multivariable Cox proportional hazards model.

A logistic regression analysis was conducted to estimate the probability of 30-day mortality, with the logit transformation (logit(p)) calculated using the coefficients of the final model. The predicted logit(p) values were converted into probabilities of 30-day mortality, and model fit was evaluated using the Hosmer–Lemeshow test.

## 3. Results

### 3.1. Flowchart and Baseline Characteristics

Figure 1 shows a flowchart of the study population. Initially, 358 patients who met diagnostic criteria for sepsis were screened by ED physicians on duty. Among them, 115 patients were excluded: refusal to participate in the study (*n* = 105), visit for trauma care (*n* = 7), and missing data in PCT levels (*n* = 3). Finally, 243 patients with sepsis were enrolled in the present study. The patients were classified into two groups: (1) sepsis (without shock) (*n* = 124), and (2) septic shock (*n* = 119). Each group was further divided into 30-day survival and mortality groups.

The baseline characteristics of the study population are presented in Table 1. There were no differences in age, sex, comorbidities, or infection sites between the two groups. Except for MEWS, clinical severity scores were higher in patients with septic shock than in those with sepsis. Systolic blood pressure, diastolic blood pressure, mean arterial pressure, and peripheral capillary oxygen saturation were lower in the septic shock group than in the non-septic shock group. Platelet and creatinine levels differed between the two groups. The WRS, P-SEP, PCT, and lactate levels were higher in patients with septic shock than in those with sepsis. CRP levels did not differ between the two groups. Mechanical ventilation was more frequently used in the septic shock group. The incidence of acute kidney injury (AKI) was higher in the septic shock group. Overall, piperacillin/tazobactam was the most frequently administered antibiotics, followed by ceftriaxone, levofloxacin, and meropenem. Overall, the 7-, 14-, and 30-day mortality rates were higher in the septic shock group.

### 3.2. Diagnostic and Prognostic Value of Biomarkers

We assessed the clinical value of WRS in patients with sepsis using ROC curve analysis (Figure 2A,B). The AUC to discriminate septic shock from sepsis was 0.588 for WRS (95% CI, 0.516–0.660; sensitivity, 67.2%; specificity, 50.8%; *p* = 0.018), 0.624 for P-SEP (95% CI, 0.554–0.695; sensitivity, 45.4%; specificity, 82.3%; *p* = 0.001), 0.685 for PCT (95% CI, 0.618–0.751; sensitivity, 59.7%; specificity, 71.0%; *p* < 0.001), and 0.669 for lactate (95% CI, 0.602–0.737; sensitivity, 93.3%; specificity, 13.7%; *p* < 0.001). The optimal cut-off values to discriminate septic shock from sepsis were 61.20 µg/L for WRS, 1286 ng/L for P-SEP, 3.08 µg/L for PCT, and 2.08 mmol/L for lactate. CRP could not discriminate between septic shock and sepsis (AUC, 0.549; 95% CI, 0.477–0.621; sensitivity, 74.8%; specificity, 23.4%; *p* = 0.187).

The AUC to predict 30-day mortality was 0.648 for WRS (95% CI, 0.569–0.726; sensitivity, 56.7%; specificity, 73.3%; *p* < 0.001), 0.610 for P-SEP (95% CI, 0.531–0.689; sensitivity, 64.2%; specificity, 57.4%; *p* = 0.008), 0.683 for lactate (95% CI, 0.606–0.759; sensitivity, 88.1%; specificity, 28.4%; *p* < 0.001). The optimal cut-off values for predicting 30-day mortality were 84.15 µg/L for WRS, 823 ng/L for P-SEP, and 4.42 mmol/L for lactate. However, CRP and PCT levels did not predict the 30-day mortality.

### 3.3. Combination of Clinical Severity Scores with Biomarkers

We investigated the prognostic value of clinical severity scores (Figure 2C). Among the four clinical severity scores, SOFA score had the highest AUC (AUC, 0.732; 95% CI, 0.662–0.802; *p* < 0.001), followed by APACHE II (AUC, 0.650; 95% CI, 0.572–0.727; *p* < 0.001), NEWS (AUC, 0.607; 95% CI, 0.523–0.691; *p* = 0.010), and MEWS (AUC, 0.553; 95% CI, 0.469–0.636; *p* = 0.206).

Figure 2D shows the multivariable logistic regression model for predicting 30-day mortality using the SOFA score and the three biomarkers (WRS, P-SEP, and lactate). The log of probability was converted into the 30-day mortality probability using a regression equation. In the ROC curve analysis, the AUC of the SOFA score alone was 0.732 (95% CI, 0.662–0.802; *p* < 0.001), and the model was well calibrated (Hosmer–Lemeshow test; chi-squared = 4.633; df = 8; *p* = 0.796). When combining the SOFA score with WRS, the AUC was 0.748 (95% CI, 0.680–0.816; *p* < 0.001), and the model was well calibrated (Hosmer–Lemeshow test; chi-square = 4.633; df = 8; *p* = 0.796). When the SOFA score was combined with three biomarkers (WRS, P-SEP, and lactate), the AUC for predicting 30-day mortality increased to 0.761 (95% CI, 0.692–0.830; *p* < 0.001), and the model showed good calibration (Hosmer–Lemeshow test; chi-squared = 10.617; df = 8; *p* = 0.224).

### 3.4. Prognostic Value of WRS According to Microbial Culture Test Results

Figure 3 shows the prognostic values of WRS, P-SEP, and lactate according to the microbial culture test results, including blood, sputum, and urine cultures. Each Kaplan–Meier survival curve was stratified by the optimal cut-off value of WRS (84.15 µg/L, Figure 3A), P-SEP (823 ng/L, Figure 3E), or lactate (4.42 mmol/L; Figure 3I) to predict 30-day mortality. Overall, patients whose biomarker levels were above the cut-off value had higher 30-day mortality than those whose biomarker levels were below the cut-off value. In addition, patients whose biomarker levels were above the cut-off value were stratified based on the results of blood, sputum, or urine culture tests.

Among patients whose WRS levels were above the cut-off value, 30-day mortality did not differ between those with positive and negative blood culture results (*p* = 0.061; Figure 3B). Thirty-day mortality did not differ between the patients with positive and negative sputum culture results (*p* = 0.745; Figure 3C). Thirty-day mortality was higher in patients with positive urine cultures than in those with negative cultures (*p* = 0.040; Figure 3D). Among patients whose P-SEP levels were above the cut-off value, 30-day mortality was higher in those with positive blood cultures than in those with negative results (*p* = 0.042; Figure 3F). Thirty-day mortality did not differ between the patients with positive and negative sputum culture results (*p* = 0.130; Figure 3G). Thirty-day mortality also did not differ between patients with positive and negative urine culture results (*p* = 0.354; Figure 3H). Among patients whose lactate levels were above the cut-off value, 30-day mortality did not differ between positive and negative results for blood culture (*p* = 0.116; Figure 3J), sputum culture (*p* = 0.737; Figure 3K), and urine culture (*p* = 0.444; Figure 3L).

### 3.5. Prognostic Value of WRS According to Each Component of SOFA Score

We classified patients whose WRS, P-SEP, or lactate levels were above the cut-off value using the central nervous system SOFA (GCS SOFA) score, cardiovascular SOFA (BP SOFA) score, coagulation SOFA (PLT SOFA) score, renal SOFA (Cr SOFA) score, liver SOFA (Bili SOFA) score, and respiration SOFA (Resp SOFA) score (Figure 4A–R).

Among patients whose WRS levels were over the cut-off value, 30-day mortality was higher in those with PLT SOFA score ≥ 1 than in those with PLT SOFA score of 0 (*p* = 0.036; Figure 4C). Thirty-day mortality was higher in those with a Cr SOFA score ≥ 1 than in those with a Cr SOFA score of 0 (*p* = 0.027; Figure 4D). However, 30-day mortality did not differ in between patients with the other components of SOFA score ≥ 1 and those with the other components of SOFA score of 0: GCS SOFA score (*p* = 0.655; Figure 4A), BP SOFA score (*p* = 0.109; Figure 4B), Bili SOFA score (*p* = 0.125; Figure 4E), and Resp SOFA score (*p* = 0.797; Figure 4F).

Among patients whose P-SEP levels were over the cut-off value, 30-day mortality was higher in those with a BP SOFA score ≥ 1 than in those with a BP SOFA score of 0 (*p* = 0.014; Figure 4H). Thirty-day mortality was higher in patients with a PLT SOFA score ≥ 1 than in those with a PLT SOFA score of 0 (*p* = 0.002; Figure 4I). However, classification using GCS SOFA score (*p* = 0.506; Figure 4G), Cr SOFA score (*p* = 0.472; Figure 4J), Bili SOFA score (*p* = 0.070; Figure 4K), or Resp SOFA score (*p* = 0.349; Figure 4L) did not show differences in 30-day mortality between the two groups (each component of SOFA score of ≥1 vs. that of SOFA score of 0).

Among patients whose lactate levels were over the cut-off value, the 30-day mortality was higher in patients with a SOFA score ≥ 1 than in those with a SOFA score of 0 when patients were classified by BP SOFA score (*p* < 0.001; Figure 4N), PLT SOFA score (*p* = 0.001; Figure 4O), and Bili SOFA score (*p* = 0.001; Figure 4Q). When patients were classified by GCS SOFA score (*p* = 0.968; Figure 4M), Cr SOFA score (*p* = 0.111; Figure 4P), and Resp SOFA score (*p* = 0.090; Figure 4R), 30-day mortality did not differ between patients with each component SOFA score of ≥1 and those with each component SOFA score of 0.

### 3.6. Correlation Between WRS and Clinical Variables

Spearman’s correlation analysis showed that creatinine levels positively correlated with WRS (rho = 0.183, *p* = 0.004), P-SEP (rho = 0.541, *p* < 0.001), CRP (rho = 0.218, *p* = 0.001), PCT (rho = 0.302, *p* < 0.001), and lactate (rho = 0.191, *p* = 0.003). Among the tested biomarkers, the correlation coefficient was highest for P-SEP and lowest for WRS. The WRS levels positively correlated with the SOFA scores (rho = 0.223, *p* < 0.001). However, WRS did not correlate with NEWS (rho = 0.006, *p* = 0.926), MEWS (rho = 0.032, *p* = 0.619), or APACHE II (rho = 0.112, *p* = 0.082).

### 3.7. Comparison of Biomarker Levels According to Cr SOFA Score

Figure 5 shows serum biomarker levels according to the Cr SOFA score. Among the enrolled patients, those who had pre-existing kidney disease were excluded from this analysis (*n* = 20). Using the Cr SOFA score of 0 as a reference value (*n* = 98), the patients with sepsis, including septic shock, were sequentially stratified according to Cr SOFA score. Then, we compared biomarker levels using Mann–Whitney *U*-test.

Patients with a Cr SOFA score of ≥1 (*n* = 125) did not have higher WRS levels than those in the reference group (93.35 vs. 82.79 µg/L, *p* > 0.05). However, all the other biomarkers were higher in patients with a Cr SOFA score of ≥1 than in the reference group (P-SEP, 1660 vs. 778 ng/L, *p* < 0.001; CRP, 15.10 vs. 10.94 mg/L, *p* < 0.01; PCT, 18.39 vs. 7.56 µg/L, *p* < 0.01; lactate, 5.41 vs. 3.97 mmol/L, *p* < 0.05).

Patients with a Cr SOFA score ≥ 2 (*n* = 55) did not have higher CRP levels than those in the reference group (17.40 vs. 10.94 mg/L, *p* > 0.05). However, the other biomarkers were higher in patients with Cr SOFA ≥ 2 than in the reference group (WRS, 120.61 vs. 82.79 µg/L, *p* < 0.05; P-SEP, 2371 vs. 778 ng/L, *p* < 0.001; PCT, 25.47 vs. 7.56 µg/L, *p* < 0.001; lactate, 6.68 vs. 3.97 mmol/L, *p* < 0.001).

Patients with a Cr SOFA score ≥ 3 (*n* = 17) did not have higher WRS levels than the reference group (104.43 vs. 82.79 µg/L, *p* > 0.05). However, the other biomarkers were higher in patients with Cr SOFA ≥ 2 than in the reference group (P-SEP, 2623 vs. 778 ng/L, *p* < 0.001; CRP, 17.82 vs. 10.94 mg/L, *p* < 0.05; PCT, 24.61 vs. 7.56 µg/L, *p* < 0.01; lactate, 7.11 vs. 3.97 mmol/L, *p* < 0.05).

Patients with Cr SOFA score of 4 (*n* = 6) did not have higher WRS or CRP levels than the reference group (WRS, 93.89 vs. 82.79 µg/L, *p* > 0.05; CRP, 15.98 vs. 10.94 mg/L, *p* > 0.05). However, P-SEP, PCT, and lactate levels were higher in patients with Cr SOFA score of 4 than in the reference group (P-SEP, 3088 vs. 778 ng/L, *p* < 0.001; PCT, 43.17 vs. 7.56 µg/L, *p* < 0.05; lactate, 6.62 vs. 3.97 mmol/L, *p* < 0.05).

### 3.8. Risk Factors for 30-Day Mortality in Sepsis

Cox proportional hazards model analysis was conducted to determine the risk factors associated with 30-day mortality among patients with sepsis (Table 2). In the univariable analysis, the risk factors for 30-day mortality were WRS (hazard ratio [HR], 1.003; 95% CI, 1.001–1.005; *p* = 0.002) and lactate (HR, 1.116; 95% CI, 1.075–1.160; *p* < 0.001) levels. In the multivariable analysis, the risk factors for 30-day mortality were WRS (HR, 1.003; 95% CI, 1.001–1.005; *p* = 0.014) and lactate (HR, 1.110; 95% CI, 1.068–1.154; *p* < 0.001) levels. P-SEP, CRP, and PCT levels did not predict the 30-day mortality.

## 4. Discussion

We investigated the clinical value of WRS in sepsis patients who visited the ED. Those experiencing septic shock demonstrated greater WRS levels relative to patients with sepsis. WRS can also effectively predict 30-day mortality. Notably, WRS demonstrated strong predictive performance for 30-day mortality among sepsis patients presenting with bacteriuria or increased serum creatinine. A comparison of biomarker levels according to the Cr SOFA score suggests that WRS levels might be less influenced by kidney dysfunction than the established biomarkers.

According to the Sepsis-3 definition, sepsis is a severe medical condition characterized by organ failure triggered by a dysregulated host response to infection [1]. The performance of the SOFA score in predicting in-hospital mortality was similar to that of the Logistic Organ Dysfunction System and better than that of the SIRS or quick SOFA (qSOFA) [8]. The SOFA score is now widely used to evaluate the clinical severity of organ dysfunction [12,13,14]. A recent study showed that WRS levels positively correlated with SOFA score and mortality [19]. Similarly to the previous study, our results showed that WRS levels positively correlated with the SOFA score and creatinine, P-SEP, CRP, PCT, and lactate levels in patients with sepsis. These findings suggest that WRS levels could be associated with both organ dysfunction and prognosis in critically ill patients.

A recent ICU-based study reported that WRS could be useful not only for discriminating sepsis from non-infectious inflammation but also for predicting 28-day mortality in sepsis [25]. In that study, the AUC of WRS to predict 28-day mortality was 0.687 (cut-off value, 97.23 µg/L) in patients with sepsis. Similarly, our study showed that the AUC of WRS to predict 30-day mortality was 0.648 (optimal cut-off value, 84.15 µg/L) in patients with sepsis. Our results suggest that WRS levels can help predict short-term mortality among patients with sepsis in the ED.

AKI is a critical and common sepsis complication. Up to 60% of patients with sepsis experience AKI during their hospital stay [27,28]. Among patients with sepsis, those without pre-existing kidney disease should be considered a high-risk group for developing AKI [29]. Older adults are more vulnerable to urosepsis because one of the most common infection sources in this age group is the urinary tract [30]. PCT performs well in diagnosing sepsis, but its prognostic value for sepsis or septic shock is limited [16,31,32]. P-SEP may be a useful biomarker in the ED because plasma P-SEP levels are more rapidly elevated than those of PCT or CRP in patients with sepsis [33]. However, P-SEP has limited diagnostic performance in patients with kidney dysfunction because P-SEP levels positively correlate with Cr levels [34,35].

Our study showed that WRS has excellent prognostic value in sepsis patients with positive urine culture results or high serum creatinine levels. This result suggests that patients with elevated WRS levels and positive urine culture or high serum creatinine levels should be considered as a high-risk group for mortality and treated more carefully. For these patients, we recommend the earlier initiation of active interventions, including renal replacement therapy and administration of antibiotics.

Our study showed that WRS had the lowest correlation coefficient with serum creatinine levels among the tested biomarkers. If the biomarker levels are strongly influenced by kidney dysfunction and serum creatinine levels, it may be difficult to discriminate sepsis without AKI from AKI without sepsis. For example, P-SEP levels in patients receiving hemodialysis without infection may be as high as those in patients with sepsis or septic shock [35]. WRS levels appear to be less influenced by serum creatinine levels than by other biomarkers in patients with sepsis. This suggests that WRS levels can predict sepsis more effectively than other established biomarkers in creatinine-elevated conditions.

A previous study suggested that WRS could be a therapeutic target for various clinical conditions, such as malignancy [20]. Another study that examined the expression of WRS in cancer tissues indicated that the role of WRS in tumor biology is complex and appears to be context-dependent [36]. Studies have shown that elevated WRS levels in gastric adenocarcinoma, ovarian, and colorectal cancers correlate with favorable prognosis and can serve as predictive markers to guide the avoidance of adjuvant chemotherapy post-surgery in resectable gastric adenocarcinoma [37,38,39]. In contrast, elevated WRS levels positively correlated with tumor stage, invasion, and depth in oral squamous cell carcinoma [40]. Our data showed that median WRS levels were higher in patients with malignancy than in those without malignancy (100.60 vs. 64.50 µg/L, *p* < 0.001) among patients with sepsis. Although this suggests that high levels of WRS might be associated with malignancy, the commercial kit used in our study could detect only FL-WRS, not mini-WRS. The alternative splice form Mini-WRS, which lacks the N-terminal 47 amino acids, has been shown to exert antiangiogenic effects [41,42]. However, FL-WRS, which retains the entire N-terminal extension, does not display angiostatic activity [43,44]. Since the commercial assay exclusively detects FL-WRS, the reason for the markedly elevated FL-WRS levels observed in patients with malignancy in our study remains unclear. Additional experimental investigations are warranted to elucidate the role of WRS in tumor biology.

Our work had several limitations. First, this was a single-center ED-based study, making the generalization of our results to other populations difficult. A prospective multicenter study with a bigger sample size should be performed to confirm our results. Second, we did not analyze the serial measurements of WRS levels because our study only measured the initial WRS levels in the ED. Further prospective studies with serial measurements of WRS levels are needed to reflect the dynamic changes in patients with sepsis and septic shock. Third, there was a time gap between blood sampling and WRS measurements. Venous blood was drawn into ethylenediaminetetraacetic acid tubes within 6 h following admission to the emergency department. To minimize protein degradation, blood samples were centrifuged at 800× *g* for 15 min at 4 °C. The plasma supernatants were then aliquoted without delay and stored at −80 °C in the biobank. All analyses were performed within 3 months of blood sampling in the ED. We defrosted the blood sample only once for the WRS measurement. Fourth, this study did not include a control group, such as patients with non-infectious inflammation or non-infectious organ failure NIOF. Further prospective studies including appropriate control groups are required to evaluate the diagnostic performance of the WRS among critically ill patients. Finally, our study did not investigate the effect of dialyzer or peritoneal clearance on WRS levels. WRS levels may be influenced by renal replacement therapy. Further experimental studies are required to estimate the clearance of WRS among sepsis patients receiving the intervention

## 5. Conclusions

WRS is a novel biomarker that can help predict clinical outcomes in patients having sepsis. In particular, the prognostic performance of WRS levels was remarkable in sepsis patients with kidney dysfunction or urinary tract infection. The combination of SOFA score, WRS, P-SEP, and lactate level showed excellent value in predicting 30-day mortality among patients with sepsis. Future large-scale, prospective multicenter investigations are required to confirm these results.

## Figures and Tables

**Figure 1 diagnostics-15-02634-f001:**
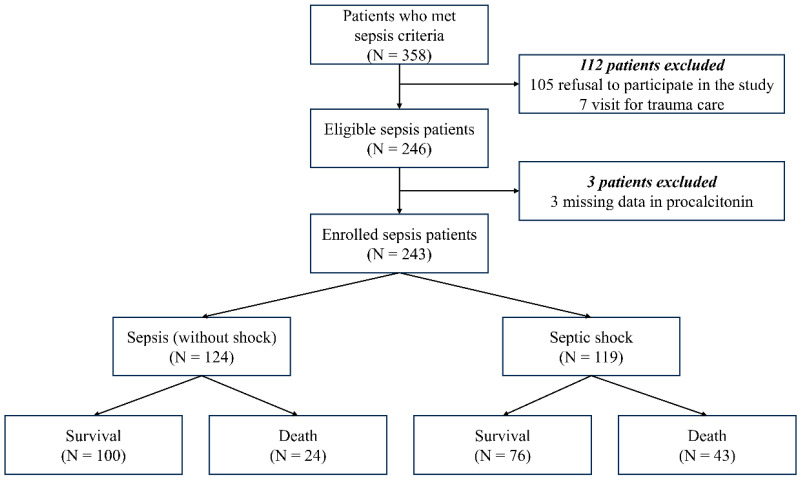
Flowchart of the study population.

**Figure 2 diagnostics-15-02634-f002:**
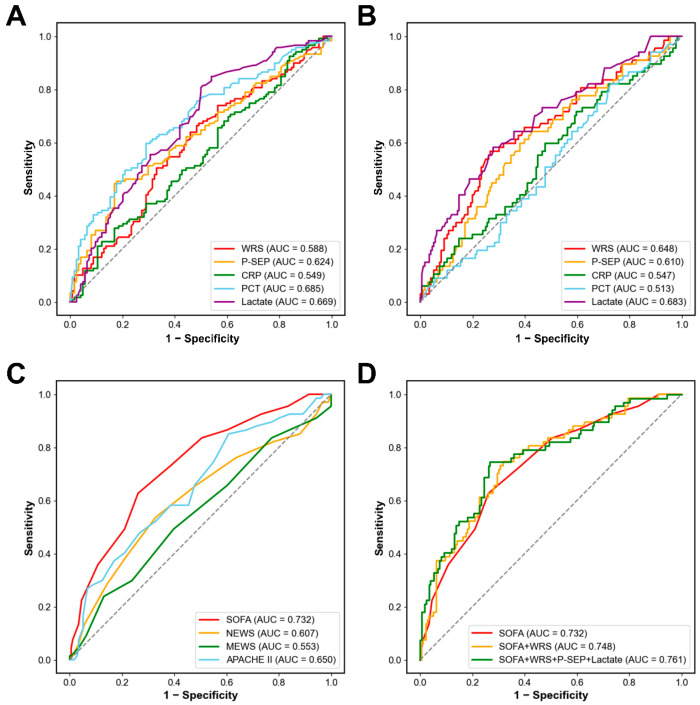
Receiver operating characteristic curves analysis. (**A**) Diagnostic value of biomarkers for discriminating septic shock from sepsis. (**B**) Prognostic value of biomarkers for predicting 30-day mortality. (**C**) Prognostic value of early warning scores or severity indices for predicting 30-day mortality. (**D**) Combinations of the SOFA score with biomarkers for predicting 30-day mortality. WRS, tryptophanyl-tRNA synthetase; P-SEP, presepsin; CRP, C-reactive protein; PCT, procalcitonin; SOFA, sequential organ failure assessment; NEWS, national early warning score; MEWS, modified early warning score; APACHE II, acute physiology and chronic health evaluation II; AUC, area under the curve.

**Figure 3 diagnostics-15-02634-f003:**
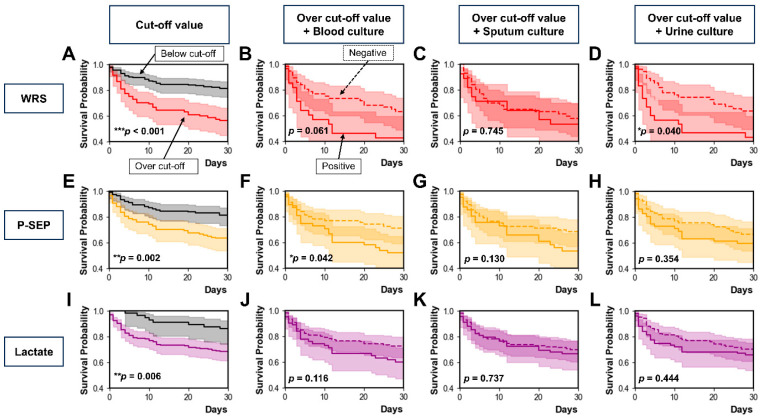
Kaplan–Meier survival curves analysis according to microorganism culture test results. (**A**–**D**) WRS. (**E**–**H**) P-SEP. (**I**–**L**) Lactate. (**A**,**E**,**I**) Biomarker levels over the cut-off value vs. below the cut-off value. (**B**,**F**,**J**) Biomarker levels over the cut-off value with blood culture positive vs. negative. (**C**,**G**,**K**) Biomarker levels over the cut-off value with sputum culture positive vs. negative. (**D**,**H**,**L**) Biomarker levels over the cut-off value with urine culture positive vs. negative. * *p* < 0.05, ** *p* < 0.01, *** *p* < 0.001. WRS, tryptophanyl-tRNA synthetase; P-SEP, presepsin.

**Figure 4 diagnostics-15-02634-f004:**
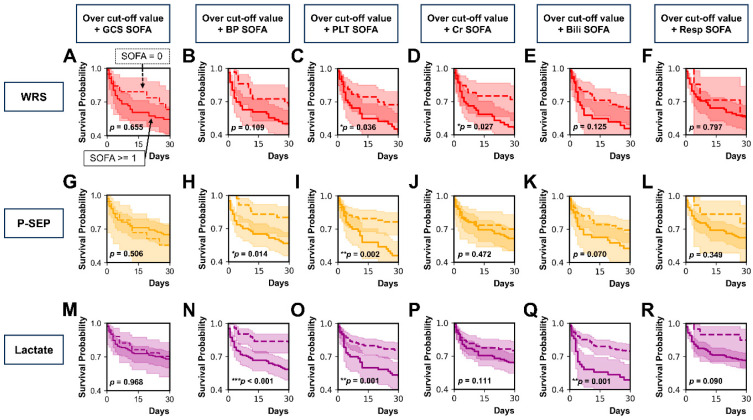
Kaplan–Meier survival curves analysis according to SOFA score. (**A**–**F**) WRS. (**G**–**L**) P-SEP. (**M**–**R**) Lactate. (**A**,**G**,**M**) GCS SOFA score ≥ 1 vs. GCS SOFA score of 0. (**B**,**H**,**N**) BP SOFA score ≥ 1 vs. BP SOFA score of 0. (**C**,**I**,**O**) PLT SOFA score ≥ 1 vs. PLT SOFA score of 0. (**D**,**J**,**P**) Cr SOFA score ≥ 1 vs. Cr SOFA score of 0. (**E**,**K**,**Q**) Bili SOFA score ≥ 1 vs. Bili SOFA score of 0. (**F**,**L**,**R**) Resp SOFA score ≥ 1 vs. GCS SOFA score of 0. * *p* < 0.05, ** *p* < 0.01, *** *p* < 0.001. WRS, tryptophanyl-tRNA synthetase; P-SEP, presepsin; GCS SOFA, central nervous system SOFA; BP SOFA, cardiovascular SOFA; PLT SOFA, coagulation SOFA; Cr SOFA, renal SOFA score; Bili SOFA, liver SOFA; Resp SOFA, respiration SOFA.

**Figure 5 diagnostics-15-02634-f005:**
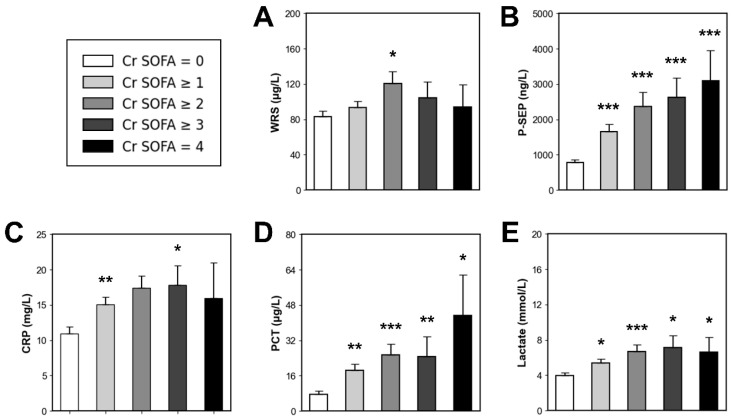
Comparison of serum biomarker levels according to Cr SOFA score. (**A**) WRS. (**B**) P-SEP. (**C**) CRP. (**D**) PCT. (**E**) Lactate. * *p* < 0.05, ** *p* < 0.01, *** *p* < 0.001. WRS, tryptophanyl-tRNA synthetase; P-SEP, presepsin; CRP, C-reactive protein; PCT, procalcitonin; Cr SOFA, renal SOFA.

**Table 1 diagnostics-15-02634-t001:** Baseline characteristics of the study population.

Variable	All Patients (*n* = 243)	Sepsis (*n* = 124)	Septic Shock (*n* = 119)	*p*-Value
**Demographics**				
Age, median (IQR)	78 (67–84)	78 (69–84)	78 (67–84)	0.973
Male, *n* (%)	144 (59.3)	77 (62.1)	67 (56.3)	0.358
**Comorbidity, *n* (%)**				
Diabetes mellitus	90 (37.0)	45 (36.3)	45 (37.8)	0.806
Hypertension	127 (52.3)	70 (56.5)	57 (47.9)	0.182
Malignancy	40 (16.5)	24 (19.4)	16 (13.4)	0.214
Chronic lung disease	20 (8.2)	7 (5.6)	13 (10.3)	0.134
Chronic liver disease	10 (4.1)	3 (2.4)	7 (5.9)	0.209
Chronic kidney disease	20 (8.2)	9 (7.3)	11 (9.2)	0.573
Cardiovascular disease	23 (9.5)	10 (8.1)	13 (10.9)	0.446
Cerebrovascular disease	56 (23.0)	28 (22.6)	28 (23.5)	0.861
**Infection sites, *n* (%)**				
Respiratory	163 (67.1)	87 (70.2)	76 (63.9)	0.296
Genitourinary	79 (32.5)	34 (27.4)	45 (37.8)	0.084
Gastrointestinal	18 (7.4)	8 (6.5)	10 (8.4)	0.561
Others	17 (7.0)	10 (8.1)	7 (5.9)	0.505
**Vital sign, median (IQR)**				
SBP (mmHg)	96 (84–122)	104 (92–136)	89 (77–102)	<0.001
DBP (mmHg)	59 (50–71)	63 (56–74)	53 (48–65)	<0.001
MAP (mmHg)	72 (62–87)	75 (70–96)	65 (57–78)	<0.001
Heart rate (bpm)	108 (88–124)	108 (88–125)	107 (88–125)	0.639
Respiratory rate (breaths/min)	24 (20–28)	23 (20–26)	24 (20–28)	0.663
Body temperature (°C)	37.2 (36.3–38.2)	37.4 (36.6–38.4)	37.2 (36.2–38.1)	0.028
SpO_2_ (%)	95 (90–98)	96 (93–99)	93 (88–97)	<0.001
**Clinical scores, median (IQR)**				
SOFA score	8 (6–11)	6 (5–8)	10 (8–12)	<0.001
NEWS	11 (9–13)	10 (8–12)	11 (9–14)	0.003
MEWS	6 (5–8)	6 (5–7)	6 (5–8)	0.076
APACHE II score	27 (23–32)	26 (21–30)	28 (24–33)	0.001
**Laboratory results, median (IQR)**				
White blood cell (×10^6^/L)	11.6 (7.8–18.4)	12.7 (8.9–17.2)	11.0 (6.2–19.8)	0.343
Hemoglobin (g/dL)	10.7 (8.9–12.4)	10.7 (8.9–12.6)	10.7 (8.6–12.3)	0.708
Platelet (×10^6^/L)	199 (120–290)	207 (141–303)	188 (104–263)	0.010
Total bilirubin (mg/dL)	0.7 (0.4–1.2)	0.7 (0.4–1.2)	0.7 (0.5–1.2)	0.749
Creatinine (mg/dL)	1.4 (0.9–2.2)	1.2 (0.9–1.9)	1.6 (1.0–2.5)	0.007
Sodium (mmol/L)	138 (133–141)	137 (133–141)	138 (133–141)	0.840
Potassium (mmol/L)	4.2 (3.7–4.8)	4.1 (3.7–4.7)	4.3 (3.6–5.1)	0.430
**Biomarkers, median (IQR)**				
WRS (µg/L)	66.70 (46.10–105.60)	60.95 (43.80–101.38)	76.30 (52.60–111.40)	0.018
P-SEP (ng/L)	804 (450–1612)	693 (412–1080)	1002 (515–2170)	<0.001
CRP (mg/L)	10.72 (4.79–17.84)	10.54 (4.51–17.10)	11.58 (5.19–21.46)	0.187
PCT (µg/L)	2.31 (0.64–10.27)	1.04 (0.41–4.79)	4.92 (1.11–23.85)	<0.001
Lactate (mmol/L)	3.00 (1.90–6.00)	2.35 (1.52–4.71)	4.10 (2.40–7.60)	<0.001
**Mechanical ventilator, *n* (%)**	70 (28.8)	15 (12.1)	55 (46.2)	<0.001
**AKI, *n* (%)**	122 (50.2)	53 (42.7)	69 (58.0)	0.018
**Renal replacement therapy, *n* (%)**				
Patients with AKI in ED	20 (8.2)	5 (4.0)	15 (12.6)	0.015
Patients without AKI in ED	8 (3.3)	2 (1.6)	6 (5.0)	0.135
**Antibiotics, *n* (%)**				
Piperacillin/Tazobactam	173 (71.2)	75 (60.5)	98 (82.4)	0.054
Ceftriaxone	63 (25.9)	42 (33.9)	21 (17.6)	0.001
Levofloxacin	16 (6.6)	6 (4.8)	10 (8.4)	0.318
Meropenem	12 (4.9)	3 (2.4)	9 (7.6)	0.100
Azithromycin	11 (4.5)	9 (7.3)	2 (1.7)	0.030
Metronidazole	10 (4.1)	2 (1.6)	8 (6.7)	0.056
Cefepime	9 (3.7)	6 (4.8)	3 (2.5)	0.302
Others	19 (7.8)	8 (6.5)	11 (9.2)	0.581
**Clinical outcomes, *n* (%)**				
7-day mortality	40 (16.5)	11 (8.9)	29 (24.4)	0.001
14-day mortality	54 (22.2)	18 (14.5)	36 (30.3)	0.003
30-day mortality	67 (27.6)	24 (19.4)	43 (36.1)	0.003

Statistical significance refers to the comparison between the sepsis and septic shock groups. IQR, interquartile range; SBP, systolic blood pressure; DBP, diastolic blood pressure; MAP, mean arterial pressure; SpO_2_, peripheral capillary oxygen saturation; SOFA, sepsis-related organ failure assessment; NEWS, National Early Warning Score; MEWS, Modified Early Warning Score; APACHE II, Acute Physiology and Chronic Health Evaluation II; WRS, tryptophanyl-tRNA synthetase; P-SEP, presepsin; CRP, C-reactive protein; PCT, procalcitonin; AKI, acute kidney injury; ED, emergency department.

**Table 2 diagnostics-15-02634-t002:** Risk factors for 30-day mortality among patients with sepsis using the Cox proportional hazards model.

Variable	Univariable HR (95% CI)	*p*-Value	Multivariable HR (95% CI)	*p*-Value
WRS	1.003 (1.001–1.005)	0.002	1.003 (1.001–1.005)	0.014
P-SEP	1.000 (1.000–1.000)	0.091		
CRP	1.015 (0.995–1.037)	0.148		
PCT	1.002 (0.992–1.011)	0.694		
Lactate	1.116 (1.075–1.160)	<0.001	1.110 (1.068–1.154)	<0.001

WRS, tryptophanyl-tRNA synthetase; P-SEP, presepsin; CRP, C-reactive protein; PCT, procalcitonin; HR, hazard ratio; CI, confidence interval.

## Data Availability

Raw data can be obtained upon request from the corresponding author.

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
