# Peer review of "Prognostic Value of Tryptophanyl-tRNA Synthetase in Sepsis Combined with Kidney Dysfunction or Urinary Tract Infection: A Prospective Observational Study"

_diagnostics, 2025, doi:10.3390/diagnostics15202634_

Round 1
Reviewer 1 Report
Comments and Suggestions for Authors
This is an interesting prospective observational study regarding the prognostic value of WRS in patients with sepsis. There are several concerns that must be addressed by the authors.
Major revisions
The number of the patients with acute kidney injury treated with renal replacement therapies should be reported. In this respect, the effect of dialyzer or the peritoneal clearance of WRS should be investigated.
In addition, the cost of this new biomarker in the clinical practice should be also reported.
Author Response
This is an interesting prospective observational study regarding the prognostic value of WRS in patients with sepsis. There are several concerns that must be addressed by the authors.
Major revisions
Comments 1: The number of the patients with acute kidney injury treated with renal replacement therapies should be reported. In this respect, the effect of dialyzer or the peritoneal clearance of WRS should be investigated.
Response 1: Thank you for your valuable comments. Among 243 patients enrolled in the present study, 122 patients met the criteria for acute kidney injury. Among them, 20 patients were treated with renal replacement therapy (RRT). We have provided this information in Table 1. To the best of our knowledge, no study has investigated the effect of dialyzer or the peritoneal clearance on WRS levels. Because our study aimed to investigate the prognostic value of WRS among patients with sepsis, data regarding the effect of RRT could not be analyzed. Our study analyzed only initial WRS levels measured on ED presentation (no subsequent measurement of WRS), thus could not estimate the effect of hemodialysis or peritoneal dialysis on WRS levels. According to previous studies, the molecular weight of full-length WRS is 50–60 kDa [1-3], which is bigger than that of procalcitonin but smaller than that of CRP. In accordance with your recommendations, further experimental studies are required to estimate the clearance of WRS among sepsis patients receiving RRT. We have provided this point in our revised manuscript.
- Johnson, J. D., Spellman, J. M., White, K. H., Barr, K. K., & John, T. R. (2002). Human tryptophanyl-tRNA synthetase can efficiently complement the Saccharoymces cerevisiae homologue, Wrs1P. FEMS microbiology letters, 216(1), 111–115. https://doi.org/10.1111/j.1574-6968.2002.tb11423.x
- Shen, N., Guo, L., Yang, B., Jin, Y., & Ding, J. (2006). Structure of human tryptophanyl-tRNA synthetase in complex with tRNATrp reveals the molecular basis of tRNA recognition and specificity.Nucleic acids research,34(11), 3246–3258. https://doi.org/10.1093/nar/gkl441
- Ahn, Y. H., Oh, S. C., Zhou, S., & Kim, T. D. (2021). Tryptophanyl-tRNA Synthetase as a Potential Therapeutic Target.International journal of molecular sciences,22(9), 4523. https://doi.org/10.3390/ijms22094523
Comments 2: In addition, the cost of this new biomarker in the clinical practice should be also reported.
Response 2: Thank you for your valuable comment. In the 3rd paragraph of the introduction, we have provided the estimated cost of the single WRS levels measurement. According to the development company, estimated cost after commercialization would be approximately $10–20 per each measurement.
Reviewer 2 Report
Comments and Suggestions for Authors
Thank you for the opportunity to review this article, which addresses one of the most pressing and critical topics in emergency medicine: the diagnosis and prognosis of outcomes in sepsis associated with renal impairment. The authors propose using the Tryptophanyl-tRNA synthetase (WRS) test as one of the sepsis indicators. The search for such criteria is crucial, as sepsis can occur in the context of severe immunodeficiency or hematological diseases, when we cannot use inflammatory markers, leukocytosis, C-reactive protein, etc. as sepsis criteria. Therefore, I believe the Introduction section should provide a more detailed description of the WRS test proposed by the authors, including its availability, sensitivity, and specificity, according to literature data, its cost, and whether it can be used universally. In the Materials and Methods chapter, it is necessary to indicate the type of study design (cross-sectional?) and clearly state the inclusion and exclusion criteria for the study.
The Materials and Methods chapter should specify the study design (cross-sectional?) and clearly outline the inclusion and exclusion criteria.
It is important to specify the treatment received by the patients included in the study, including the antibiotics used, as the use of antibacterial drugs can reduce kidney function due to nephrotoxicity.
Figure 2 and Table 2 duplicate information. Furthermore, the authors provide these same figures in great detail in the text of the manuscript. This is unnecessary. This duplication of figures makes the text difficult to understand.
In the Discussion section, the authors, for unknown reasons, did not provide a comparison of their results with data from other authors who studied the effectiveness of the tryptophanyl-tRNA synthetase test in sepsis. This information should be added to the manuscript.
Author Response
Thank you for the opportunity to review this article, which addresses one of the most pressing and critical topics in emergency medicine: the diagnosis and prognosis of outcomes in sepsis associated with renal impairment. The authors propose using the Tryptophanyl-tRNA synthetase (WRS) test as one of the sepsis indicators. The search for such criteria is crucial, as sepsis can occur in the context of severe immunodeficiency or hematological diseases, when we cannot use inflammatory markers, leukocytosis, C-reactive protein, etc. as sepsis criteria.
Comments 1: Therefore, I believe the Introduction section should provide a more detailed description of the WRS test proposed by the authors, including its availability, sensitivity, and specificity, according to literature data, its cost, and whether it can be used universally.
Response 1: Thank you for your valuable comments. According to your comments, we have provided the availability, sensitivity, specificity, and measurement cost of WRS in the last paragraph of the introduction section, citing a recent clinical study. According to the development company, estimated cost after commercialization would be approximately $10–20 per each measurement. We have provided this information in the introduction section.
Comments 2: In the Materials and Methods chapter, it is necessary to indicate the type of study design (cross-sectional?) and clearly state the inclusion and exclusion criteria for the study. The Materials and Methods chapter should specify the study design (cross-sectional?) and clearly outline the inclusion and exclusion criteria.
Response 2: Thank you for your valuable comments. Our study design is not a cross-sectional study, but a prospective observational one. To clearly demonstrate that our research is a prospective observational study, we have described overall enrollment process and thus have updated the flowchart of the study population (Figure 1). To specify the design of our study, we also have provided the inclusion and exclusion process in both results and methods section of our revised manuscript.
Comments 3: It is important to specify the treatment received by the patients included in the study, including the antibiotics used, as the use of antibacterial drugs can reduce kidney function due to nephrotoxicity.
Response 3: Thank you for your valuable comments. According to your advice, we have provided the list of antibiotics administered to the enrolled patients in Table 1.
Comments 4: Figure 2 and Table 2 duplicate information. Furthermore, the authors provide these same figures in great detail in the text of the manuscript. This is unnecessary. This duplication of figures makes the text difficult to understand.
Response 4: Thank you for your valuable comments. We totally agree with your opinion that Figure 2 duplicated Table 2. According to your advice, we have removed Table 2 in the revised manuscript. We appreciate your thoughtful advice.
Comments 5: In the Discussion section, the authors, for unknown reasons, did not provide a comparison of their results with data from other authors who studied the effectiveness of the tryptophanyl-tRNA synthetase test in sepsis. This information should be added to the manuscript.
Response 5: Thank you for your valuable comments. However, it is not the case that we did not compare our results with those of previous studies in the discussion section. In the 2nd, 3rd, and 7th paragraphs of the discussion, we have compared our findings with those of previous studies regarding the clinical value of WRS levels in sepsis patients. We really appreciate your time and advice for our work.
Round 2
Reviewer 2 Report
Comments and Suggestions for Authors
The authors have done a great job of improving the article. I am satisfied with the changes the authors have made. I propose accepting the article for publication as it is.